

# Transcriptomic evidence for involvement of reactive oxygen species in *Rhizoctonia solani* AG1 IA sclerotia maturation

Bo Liu[1,2], Haode Wang[1], Zhoujie Ma[1], Xiaotong Gai[1], Yanqiu Sun[1], Shidao He[1], Xian Liu[3], Yanfeng Wang[2], Yuanhu Xuan[4] and Zenggui Gao[1]

[1] Institute of Plant Immunology, Shenyang Agricultural University, Shenyang, Liaoning, China
[2] College of Life Sciences, Yan'an University, Yan'an, Shaanxi, China
[3] College of Bioscience and Biotechnology, Shenyang Agricultural University, Shenyang, Liaoning, China
[4] College of Plant Protection, Shenyang Agricultural University, Shenyang, Liaoning, China

## ABSTRACT

*Rhizoctonia solani* AG1 IA is a soil-borne fungal phytopathogen that can significantly harm crops resulting in economic loss. This species overwinters in grass roots and diseased plants, and produces sclerotia that infect future crops. *R. solani* AG1 IA does not produce spores; therefore, understanding the molecular mechanism of sclerotia formation is important for crop disease control. To identify the genes involved in this process for the development of disease control targets, the transcriptomes of this species were determined at three important developmental stages (mycelium, sclerotial initiation, and sclerotial maturation) using an RNA-sequencing approach. A total of 5,016, 6,433, and 5,004 differentially expressed genes (DEGs) were identified in the sclerotial initiation vs. mycelial, sclerotial maturation vs. mycelial, and sclerotial maturation vs. sclerotial initiation stages, respectively. Moreover, gene ontology (GO) and kyoto encyclopedia of genes and genomes (KEGG) analyses showed that these DEGs were enriched in diverse categories, including oxidoreductase activity, carbohydrate metabolic process, and oxidation-reduction processes. A total of 12 DEGs were further verified using reverse transcription quantitative PCR. Among the genes examined, NADPH oxidase 1 (*NOX1*) and superoxide dismutase (*SOD*) were highly induced in the stages of sclerotial initiation and maturation. In addition, the highest reactive oxygen species (ROS) production levels were detected during sclerotial initiation, and enzyme activities of NOX1, SOD, and catalase (CAT) matched with the gene expression profiles. To further evaluate the role of ROS in sclerotial formation, *R. solani* AG1 IA was treated with the CAT inhibitor aminotriazole and $H_2O_2$, resulting in the early differentiation of sclerotia. Taken together, this study provides useful information toward understanding the molecular basis of *R. solani* AG1 IA sclerotial formation and maturation, and identified the important role of ROS in these processes.

Corresponding author
Zenggui Gao,
gaozenggui@syau.edu.cn

## INTRODUCTION

*Rhizoctonia solani* is a soil-borne pathogen belonging to the class Agaricomycetes, family Ceratobasidiaceae, that causes various plant diseases and attacks crops (maize, potatoes, rice, and soybean). *R. solani* is classified into 14 anastomosis groups (AG 1–13 and AG 1 IB) (*Ogoshi, 1987*; *Priyatmojo et al., 2001*). Maize sheath blight is one of the most serious and widely distributed diseases caused by *R. solani* AG1 IA, resulting in severe yield losses in maize-cultivating areas worldwide. The sclerotium is a special structure of *R. solani* that overwinters in the soil or diseased plants, and can survive under adverse conditions (e.g., low temperature) for long periods (*Rush & Lee, 1983*; *Boland et al., 2004*). Although sheath blight disease-resistant varieties of maize are available, these varieties are limited; thus, treating maize with fungicides remains the main approach to control the spread of *R. solani* in crops (*Zhao et al., 2006*). However, the emergence of fungicide tolerance has now made it even more difficult to control this disease. Since *R. solani* AG1 IA does not produce spores, the growth and germination of sclerotial are the key to maintaining the life of *R. solani*.

The development of sclerotia was divided into three stages: initiation, development, and maturation. The initiation stage is marked by the appearance of the sclerotium with white aerial mycelia entangled around the edges of the culture medium. In the development stage, the sclerotium becomes further entangled and increases in size along with secretion of a clear or tan-colored liquid on the surface. The maturation phase involves the accumulation of melanin in epidermal cells, and internal hardening. (*Townsend & Willetts, 1954*). Substrates associated with these stages (initiation, development, and maturation) of a typical spherical sclerotium in filamentous fungi may show a differentiated structure, which can possibly insulate itself from environmental oxygen (*Georgiou et al., 2006*). One of the most unique features of the sclerotium is that it becomes inactive when the environmental conditions are not conducive for growth. In the mature stage of the sclerotium, it can survive for many years enduring extreme temperatures, desiccation, starvation, harmful irradiation, and biological degradation (*Georgiou et al., 2006*; *Liang et al., 2010*). Thus, understanding how the sclerotia form is important for crop disease control. However, the molecular mechanisms of sclerotia formation of *R. solani* have been poorly understood.

Therefore, to understand the molecular basis of sclerotia formation, we determined the transcriptomes of *R. solani* AG1-IA in the three important developmental stages (mycelium, sclerotia initiation, and sclerotia maturation) using RNA-sequencing (RNA-seq), which has been frequently applied to the discovery of the molecular mechanisms underlying pathogenic growth and development (*Egan, Schlueter & Spooner, 2012*; *Metzker, 2010*; *O'Connell et al., 2012*). Moreover, RNA-seq performs better differentially expressed gene (DEG) analysis compared to other high-throughput technologies (*Wang, Gerstein & Snyder, 2009*). Transcriptome data were collected from the mycelium (4 days of growth), initial sclerotia (5 days of growth), and mature sclerotia (7 days of growth) of *R. solani* AG1-IA. Moreover, we examined the underlying pathways involved in sclerotonium formation. In particular, we tested the hypothesis that reactive oxygen species (ROS) are involved in the process.

Reactive oxygen species arises from mitochondrial oxidative metabolism and in the cellular response to xenobiotics, cytokines, and bacterial invasion (*Ray, Huang & Tsuji, 2012*). The influence of ROS on molecular and biochemical processes as well as signal transduction pathways is well-established, which in turn affects cell proliferation and differentiation, leading to the death of fungi and other organisms (*Allen & Tresini, 2000*). ROS are more reactive than $O_2$ in its ground state or triplet state ($^3O_2$). These ROS are dioxygen molecules in their excited singlet state forms ($^1O_2$) and partially reduced forms of oxygen, including superoxide radical ion and its protonated forms $HO_2$, hydroxyl radical (HO), and hydrogen peroxide ($H_2O_2$) (*Cui, Kong & Zhang, 2011*; *Georgiou et al., 2006*; *Turrens, 2003*). Moreover, ROS play an important role in the regulation of cellular signaling pathways; for example, hyperoxidant states are the primary driving force for cell differentiation (*Fang, Hanau & Vaillancourt, 2002*; *Lara-Ortíz, Riveros-Rosas & Aguirre, 2003*). *Chet & Henis (1975)* first proposed that the formation of sclerotia requires the participation of oxygen. However, addition of a hydroxyl radical scavenger to the culture medium of *Sclerotinia sclerotiorum* and *R. solani* was shown to suppress sclerotium formation (*Georgiou, Tairis & Sotiropoulou, 2000*). Therefore, to further understand this process, we measured the production of ROS during the three stages of sclerotium formation and determined the activities of key ROS-producing and ROS-scavenging enzymes during the process.

This work should help to identify the critical genes responsible for formation of the sclerotium of *R. solani* AG1-IA in maize toward understanding the underlying molecular mechanism. The generated results should provide a scientific foundation for the development of new strategies and targets toward the prevention of corn sheath blight.

# MATERIALS AND METHODS

## Sample collection and preparation

*Rhizoctonia solani* AG1-IA were collected from the Institute of Plant Immunology at Shenyang Agricultural University. This strongly pathogenic strain was screened as previously described (*Tingting et al., 2013*) and cultivated on potato dextrose agar (PDA), at a temperature of 26 °C. The *R. solani* AG1-IA were collected from PDA at different stages of maturation: mycelium, RWF9M (4 days of growth); sclerotial initiation, RWF9SI (5 days of growth); sclerotial maturation, RWF9S (7 days of growth) (Fig. 1). *R. solani* AG1-IA obtained were stored at −80 °C. Three mycelia growth plates were used as three biological replicates for sclerotia formation and harvesting.

## RNA quantification

RNA was extracted at the three developmental stages of the sclerotium (mycelium, sclerotium initiation, and sclerotium maturation) using TaKaRa MiniBEST Universal RNA Extraction Kit (TaKaRa, Shiga, Japan). RNA quality was assessed on 1% agarose gel. PrimeScript™ RT reagent kit with gDNA Eraser (Takara, Shiga, Japan) was used to synthesize the cDNA. Total RNA was treated with DNase to remove gDNA.

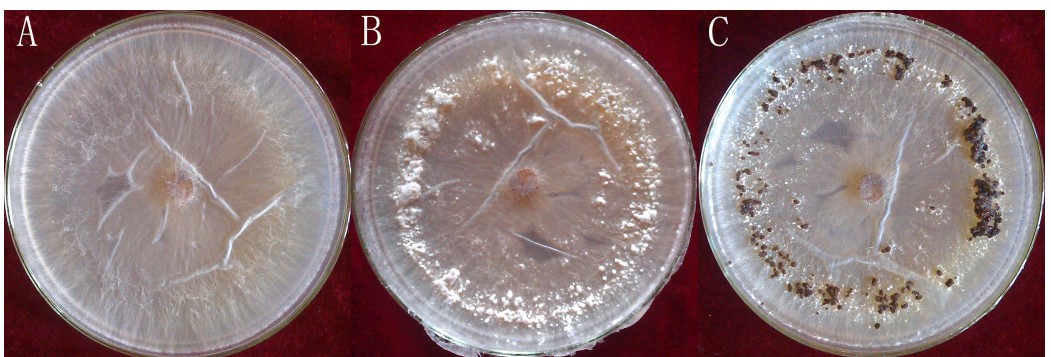

**Figure 1 Three distinct stages in the formation of sclerotia of *R. solani* AG1 IA.** (A) mycelium (four days of growth); (B) initiation (five days of growth), and (C) maturation (seven days of growth). Photo credit: Bo Liu.          

## Library preparation for transcriptome sequencing

This study provided 3 μg RNA per sample for on-machine sequencing, performed using a NEBNext® Ultra™ RNA Library Prep Kit for Illumina® (New England BioLabs, Ipswich, MA, USA). Using mRNA as a template, first-strand cDNA was synthesized using random hexamers, followed synthesize and purify second-strand cDNA. The purified second-strand cDNA was subjected to terminal repair. A tail was added, and the sequencing adapter was adaptor-ligated cDNA. Then, we produced the cDNA library by fragment size selection (150–200 bp).

## Clustering and sequencing

In the cBot Cluster Generation, the samples were clustered using the TruSeq PE Cluster Kit v3-cBot-HS (Illumina, San Diego, CA, USA); after clustering, the library was sequenced to generate 125/150 bp paired-end reads on the Illumina HiSeq platform. All raw-sequence reads were stored at Sequence Read Archive (SRP134130).

## Quality control

The raw sequence obtained by sequencing contains low-quality reads with linkers. The low-quality reads and the reads with adapter or poly-N were removed from raw data. Then, we counted the quality-control indices (Q20, Q30) and GC content (*Trapnell et al., 2010*).

## Read mapping to the reference genome

We used TopHat v2.0.12 and aligned the clean reads with those from *R. solani* AG-3 Rhs1AP genome (*Cubeta et al., 2014*; *Trapnell, Pachter & Salzberg, 2009*).

## Quantification of gene expression levels

HTSeq v0.6.1 software was used to analyze the gene expression level in each sample. The fragments per kilobase of transcript per million reads (FPKM) value of each gene was then calculated based on the length and depth (*Trapnell et al., 2010*).

## Differential expression analysis

We used the DESeq R software package (1.18.0) to analysis DEGs of the three stages (three replicates per stage) (*Wang et al., 2009*). The *P*-values of DEGs were adjusted by Benjamin

and Hochberg methods (*P*-values < 0.05) (*Anders & Huber, 2012*; *Benjamini & Hochberg, 1995*).

## GO and KEGG analysis

We used gene ontology (GO) seq R package for correcting gene (*P*-value < 0.05) and implemented GO enrichment analysis of DEGs (*Young et al., 2010*). We used KOBAS to analyze the kyoto encyclopedia of genes and genomes (KEGG) of DEGs (*Kanehisa et al., 2007*; *Mao et al., 2005*).

## Determination of ROS production

Reactive oxygen species generation in the mycelium, initial sclerotium, and mature sclerotium of *R. solani* AG1 IA was detected using dichlorodihydrofluorescein diacetate ($H_2DCFDA$) (*Ezaki et al., 2000*). Samples from the three stages were incubated with 100 μL of 10 μM $H_2DCFDA$, for 90 min. To indicate the extent of ROS production, the fluorescence staining intensity was evaluated by using a fluorescent enzyme labeling instrument SpectraMax Gemini®EM (Molecular Devices, Sunnyvale, CA, USA) at an excitation wavelength of 488 nm and emission wavelengths of 530 nm.

## Enzyme activity assays

Superoxide dismutase (SOD) enzyme activity was measured based on the method by *Beauchamp & Fridovich (1971)* with minor modifications. To 100 mg of fungi, 1 mL sodium phosphate buffer (60 mM, pH 7.0) was added. The mixture was grinded to extract SOD. The activity of the SOD enzyme was determined by adding nitroblue tetrazolium and measuring absorbance at 560 nm (*Beauchamp & Fridovich, 1971*). CAT enzyme activity was measured based on the method by *Khanam et al. (2005)* with minor modifications. To 100 mg of fungi, 1 mL buffer (16.6 mM sodium phosphate buffer (pH 7.2), 0.6 mM EDTA, and 50 mg PVP) was added and the mixture was grinded to extract CAT. The rate of consumption of hydrogen peroxide by the extract was assayed by measuring absorbance at 240 nm. Next, cell membranes were separated from fungi by two-phase partitioning (*Qiu et al., 2002*). The superoxide radicals in the membrane were determined by treatment with 0.5 mM XTT and 50 mM NADPH followed by measurement of absorbance at 470 nm. Thus, the total NOX enzyme activity was calculated (*Able, Guest & Sutherland, 1998*, *Sutherland & Learmonth, 1997*). All enzyme assays were performed in triplicates with two repetitions.

## Real-time PCR analysis

A total of 12 specific primers and four reference gene-specific primers were obtained (Table S1). The amplification efficiency of primer pairs was determined as described by *Radonić (2004)*. A total of 12 DEGs were selected for reverse transcription quantitative PCR (RT-qPCR) confirmation. The reaction mixture comprised 10 μL SYBR (TaKaRa, Shiga, Japan), 2 μL cDNA, 2 μL primer pair, and 6 μL water. The reaction was performed on a CFX-96 system (BioRad, Hercules, CA, USA). All samples were tested in triplicates with two repetitions. Mapping was performed by using SigmaPlot 12.5. We used geNorm, BestKeeper, NormFinder, and the deltaCt method to evaluate four candidate

**Table 1 Output statistics of sequencing.**

| Sample name | Raw reads | Clean reads | Clean bases | Error rate (%) | Q20 (%) | Q30 (%) | GC content (%) |
|---|---|---|---|---|---|---|---|
| RWF9M1 | 65,770,094 | 63,841,862 | 9.58G | 0.02 | 96.92 | 91.97 | 52.55 |
| RWF9M2 | 45,449,762 | 44,158,454 | 6.62G | 0.02 | 96.98 | 92.04 | 52.51 |
| RWF9M3 | 47,018,438 | 45,744,426 | 6.86G | 0.01 | 97.12 | 92.35 | 52.20 |
| RWF9SI1 | 52,198,062 | 49,959,440 | 7.49G | 0.02 | 95.92 | 90.83 | 51.73 |
| RWF9SI2 | 50,834,824 | 48,902,210 | 7.34G | 0.02 | 96.33 | 91.35 | 52.27 |
| RWF9SI3 | 49,758,464 | 47,654,054 | 7.15G | 0.02 | 95.87 | 90.79 | 51.71 |
| RWF9S1 | 54,806,008 | 53,216,446 | 7.98G | 0.02 | 96.92 | 91.94 | 51.10 |
| RWF9S2 | 52,379,712 | 50,897,498 | 7.63G | 0.02 | 96.99 | 92.08 | 51.09 |
| RWF9S3 | 59,548,520 | 57,944,112 | 8.69G | 0.01 | 97.11 | 92.33 | 51.23 |

**Notes:**
1, RWF9M1 RWF9M2 RWF9M3 three biological replicates. 2, RWF9SI1 RWF9SI2 RWF9SI3 three biological replicates. 3, RWF9S1 RWF9S2 RWF9S3 three biological replicates.

reference genes in three stages (*Andersen, Jensen & Orntoft, 2004*; *Vandesompele, 2002*; *Pfaffl et al., 2004*; *Silver et al., 2006*) (Table S2).

# RESULTS

## Transcriptome sequencing reads quality inspection and sample correlation analysis during sclerotia formation of *R. solani* AG1 IA

Total RNAs were isolated from RWF9M (mycelium), RWF9SI (sclerotial initiation), and RWF9S (sclerotial maturation) (Fig. 1). From three biological replicates, sequences of nine samples were generated to achieve 477,763,884 raw reads, ranging from 45.44 to 65.77 million reads per sample. After quality filtering, 462,318,502 clean reads remained, ranging between 44.15 and 63.84 million per sample. Based on the quality test of the clean reads (Table 1), the Q20 from all reads was between 95.87% and 97.12% while the Q30 was between 90.79% and 92.35%. These results showed that the data quality of the transcriptome was high and suitable for transcriptome analysis. The use of triplicates was employed for each biological sample on the data of original reads for the correlation test.

Prior to the DEGs analysis, the Pearson correlation coefficients between samples were determined by RNA-seq correlation analysis. As shown in Fig. 2A, the Pearson correlation coefficient of each sample was over 0.90, demonstrating good repeatability. The results of the RPKM distribution were showed in Fig. 2B and RPKM density distribution were showed in Fig. 2C.

## DEGs during sclerotia formation of *R. solani* AG1 IA

To obtain the DEGs, the gene expression of sclerotial initiation and maturation were compared to the mycelial stage. In addition, the sclerotial maturation was compared to the sclerotial initiation stage. A total of 5,016, 6,433, and 5,004 DEGs were identified in the sclerotial initiation vs. mycelial, sclerotial maturation vs. mycelial, and sclerotial maturation vs. sclerotial initiation comparisons, respectively (Fig. 3). A total of 276, 1,229, and 614 DEGs were identified in the sclerotial initiation vs. mycelial, sclerotial maturation vs. mycelial, and sclerotial maturation vs. sclerotial initiation groups, respectively (Fig. 3A).

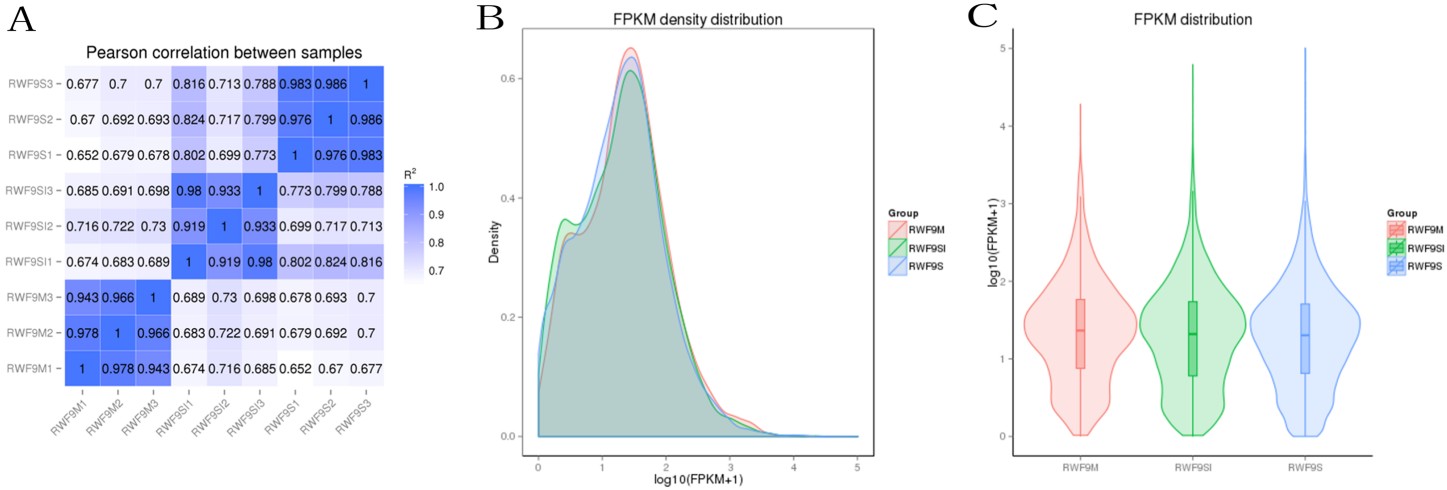

**Figure 2** **Bioinformatic analyses of RNA-seq data.** (A) Pearson correlation between samples. (B) FRKM density distribution. (C) FRKM distribution.

Overall, 2,381 and 2,635 genes were upregulated and downregulated, respectively, between the sclerotial initiation and mycelial samples. In the volcano plot, a total of 2,381, 3,102, and 2,505 DEGs were upregulated in sclerotial initiation vs. mycelial, sclerotial maturation vs. mycelial, and sclerotial maturation vs. sclerotial initiation comparisons, respectively. In contrast, a total of 2,635, 3,331, and 2,499 DEGs were downregulated in sclerotial initiation vs. mycelial, sclerotial maturation vs. mycelial, and sclerotial maturation vs. sclerotial initiation comparisons, respectively.

Under the GO terms, the unigenes were found to be involved in molecular function, cellular component, and biological process. Interestingly, the number of unigenes associated with the ribosome was particularly high in all three comparisons (Fig. 4). For the GO terms of DEGs in the sclerotial initiation vs. mycelial group, the high number of unigenes associated with single-organism metabolic, metabolic, and oxidation-reduction (Fig. 4). The sclerotial maturation vs. mycelial group were largely associated with biological process, metabolic process, oxidoreductase activity, and catalytic activity. In addition, the DEGs in the sclerotial maturation vs. sclerotial initiation group were largely associated with carbohydrate metabolic process, cellular protein metabolic process, phosphorus metabolic process, and structural molecule activity.

When the DEGs were searched against the KEGG pathway, 2,185 (sclerotial initiation vs. mycelial), 2,286 (sclerotial maturation vs. mycelial), and 1,882 (sclerotial maturation vs. sclerotial initiation) DEGs with significant hits were returned. The top 20 pathways in three groups are listed (Tables 2–4). Cluster analysis (Fig. 5) revealed that the expression pattern for DEGs in sclerotial initiation and maturation differ from that of the mycelial stage, indicating a significant change in gene expression levels during sclerotia formation. A total of eight clusters were plotted with their expression patterns (Fig. 5A). Based on the obtained FPKM data for hierarchical cluster analysis (Fig. 5B). As shown in Fig. 5, subcluster 1 included 2,334 genes with down-regulated expression at both sclerotia stages, and the expression levels of these genes were slightly lower in the sclerotial initiation and

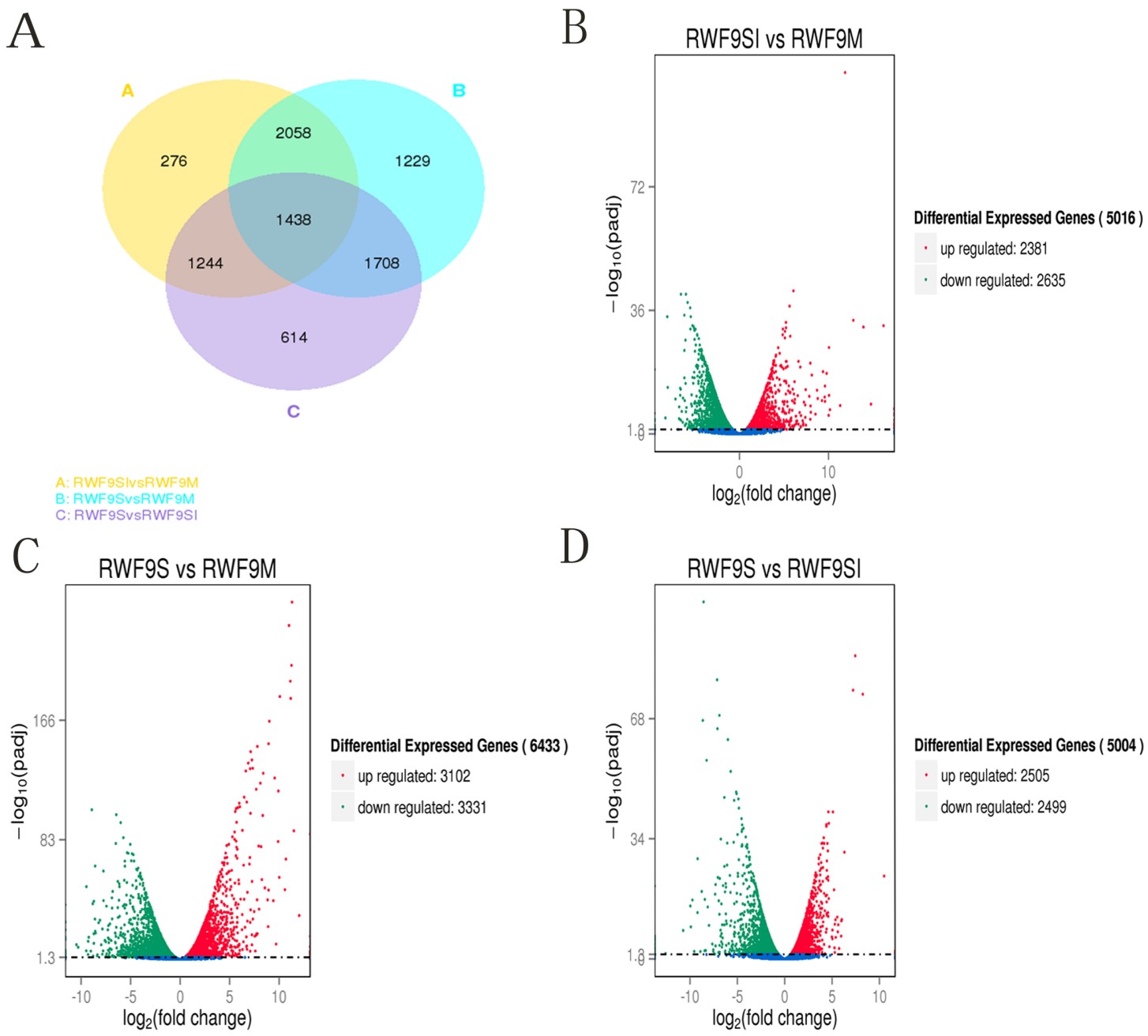

**Figure 3 Differentially expressed genes analysis.** (A) Venn diagram of differentially expressed genes in the sclerotia formation of *R. solani* AG1 IA. (B–D) Volcano plot showing genes differently expressed between different libraries. The *Q*-value for all plots was <0.005 and the absolute value of the log$^2$Ratio > 1 were used as the threshold to judge the significance of the difference in gene expression. Red points: genes up-regulate; green points: genes down-regulated; blue points: not DEGs.

maturation samples than those in the mycelial samples. The 1,396 genes in subcluster 2 were up-regulated at the sclerotial initiation stage compared to the mycelial stage, and then down-regulated at the sclerotial stage in comparison to the sclerotial initiation stage. The 220 genes in subcluster 3 were down-regulated in both sclerotial stages, the expression levels in sclerotial maturation were lower than the other two stages. Subcluster 4 included

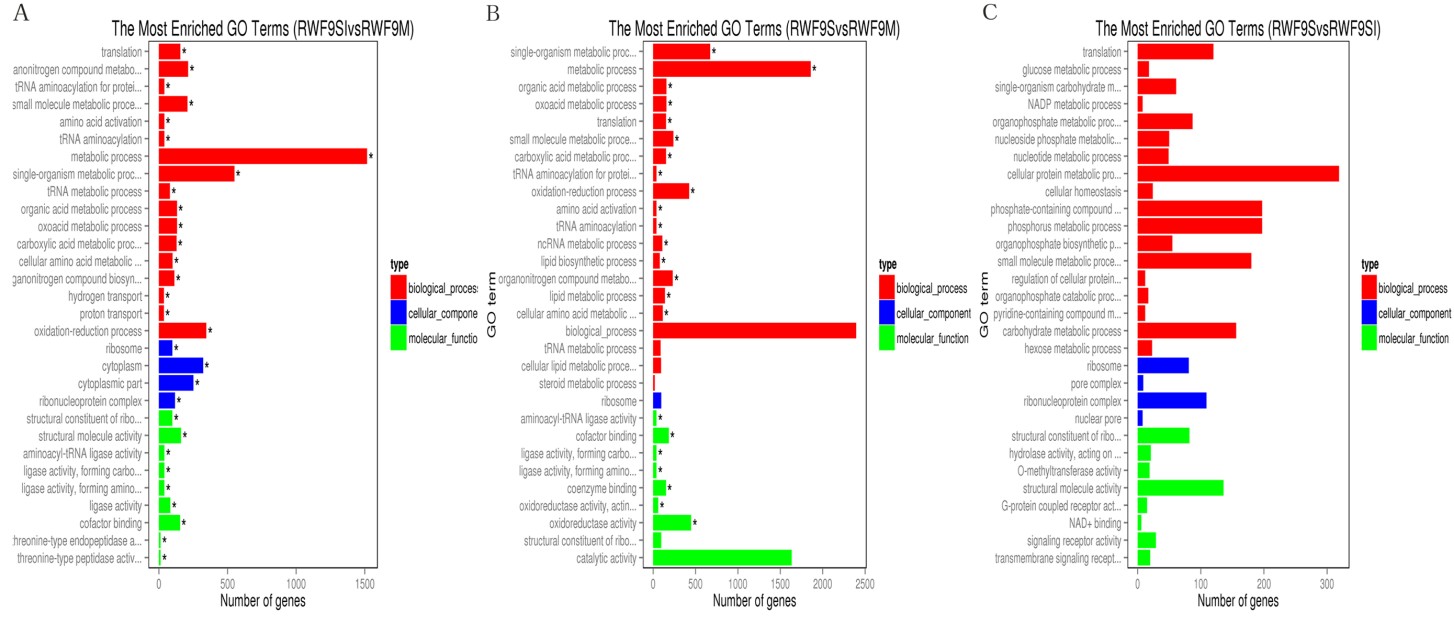

**Figure 4 The difference of GO gene enrichment column, which reflects in the biological process, cellular component, and molecular function.**
(A) The most GO enriched terms (RWF9SI vs. RWF9M). (B) The most GO enriched terms (RWF9S vs. RWF9M). (C) The most GO enriched terms
(RWF9S vs. RWF9SI). The ordinate is the enrichment of GO term; the abscissa is the number of genes in the term. Different colors distinguish
between biological processes, cell component and molecular function.

**Table 2 Kegg RWF9SI vs. RWF9M.**

| Kegg pathway | Number of unigenes | Percentage |
| --- | --- | --- |
| Metabolic pathways | 454 | 20.77 |
| Biosynthesis of secondary metabolites | 195 | 8.92 |
| Biosynthesis of amino acids | 99 | 4.53 |
| Ribosome | 91 | 4.16 |
| Carbon metabolism | 84 | 3.84 |
| RNA transport | 49 | 2.24 |
| Protein processing in endoplasmic reticulum | 44 | 2.01 |
| Oxidative phosphorylation | 43 | 1.97 |
| Spliceosome | 38 | 1.74 |
| Glycolysis/Gluconeogenesis | 36 | 1.65 |
| Cysteine and methionine metabolism | 35 | 1.6 |
| Arginine and proline metabolism | 34 | 1.56 |
| Pyruvate metabolism | 34 | 1.56 |
| 2-Oxocarboxylic acid metabolism | 32 | 1.46 |
| Proteasome | 31 | 1.42 |
| Cell cycle—yeast | 31 | 1.42 |
| mRNA surveillance pathway | 30 | 1.37 |
| Purine metabolism | 30 | 1.37 |
| Glycine, serine, and threonine metabolism | 28 | 1.28 |
| Valine, leucine, and isoleucine degradation | 26 | 1.19 |

**Table 3 Kegg RWF9S vs. RWF9M.**

| Kegg pathway | Number of unigenes | Percentage |
|---|---|---|
| Metabolic pathways | 498 | 21.78 |
| Biosynthesis of secondary metabolites | 219 | 9.58 |
| Biosynthesis of amino acids | 97 | 4.24 |
| Ribosome | 85 | 3.72 |
| Carbon metabolism | 82 | 3.59 |
| RNA transport | 51 | 2.23 |
| Protein processing in endoplasmic reticulum | 44 | 1.92 |
| Glycolysis/gluconeogenesis | 42 | 1.84 |
| Cell cycle—yeast | 39 | 1.71 |
| Oxidative phosphorylation | 38 | 1.66 |
| Pyruvate metabolism | 37 | 1.62 |
| Arginine and proline metabolism | 36 | 1.57 |
| Purine metabolism | 34 | 1.49 |
| Spliceosome | 31 | 1.36 |
| Cysteine and methionine metabolism | 30 | 1.31 |
| Glycine, serine, and threonine metabolism | 29 | 1.27 |
| Proteasome | 29 | 1.27 |
| Ribosome biogenesis in eukaryotes | 29 | 1.27 |
| Glutathione metabolism | 28 | 1.22 |
| Valine, leucine, and isoleucine degradation | 28 | 1.22 |

**Table 4 Kegg RWF9S vs. RWF9SI.**

| Kegg pathway | Number of unigenes | Percentage |
|---|---|---|
| Metabolic pathways | 342 | 18.17 |
| Biosynthesis of secondary metabolites | 134 | 7.12 |
| Ribosome | 80 | 4.25 |
| Biosynthesis of amino acids | 55 | 2.92 |
| Carbon metabolism | 49 | 2.6 |
| RNA transport | 47 | 2.5 |
| Protein processing in endoplasmic reticulum | 45 | 2.39 |
| Spliceosome | 45 | 2.39 |
| Starch and sucrose metabolism | 37 | 1.97 |
| Cell cycle—yeast | 37 | 1.97 |
| Purine metabolism | 37 | 1.97 |
| Pyrimidine metabolism | 28 | 1.49 |
| Glycolysis/Gluconeogenesis | 27 | 1.43 |
| Meiosis—yeast | 27 | 1.43 |
| mRNA surveillance pathway | 26 | 1.38 |
| Oxidative phosphorylation | 26 | 1.38 |
| Arginine and proline metabolism | 25 | 1.33 |
| Peroxisome | 25 | 1.33 |
| N-Glycan biosynthesis | 23 | 1.22 |
| Amino sugar and nucleotide sugar metabolism | 23 | 1.22 |

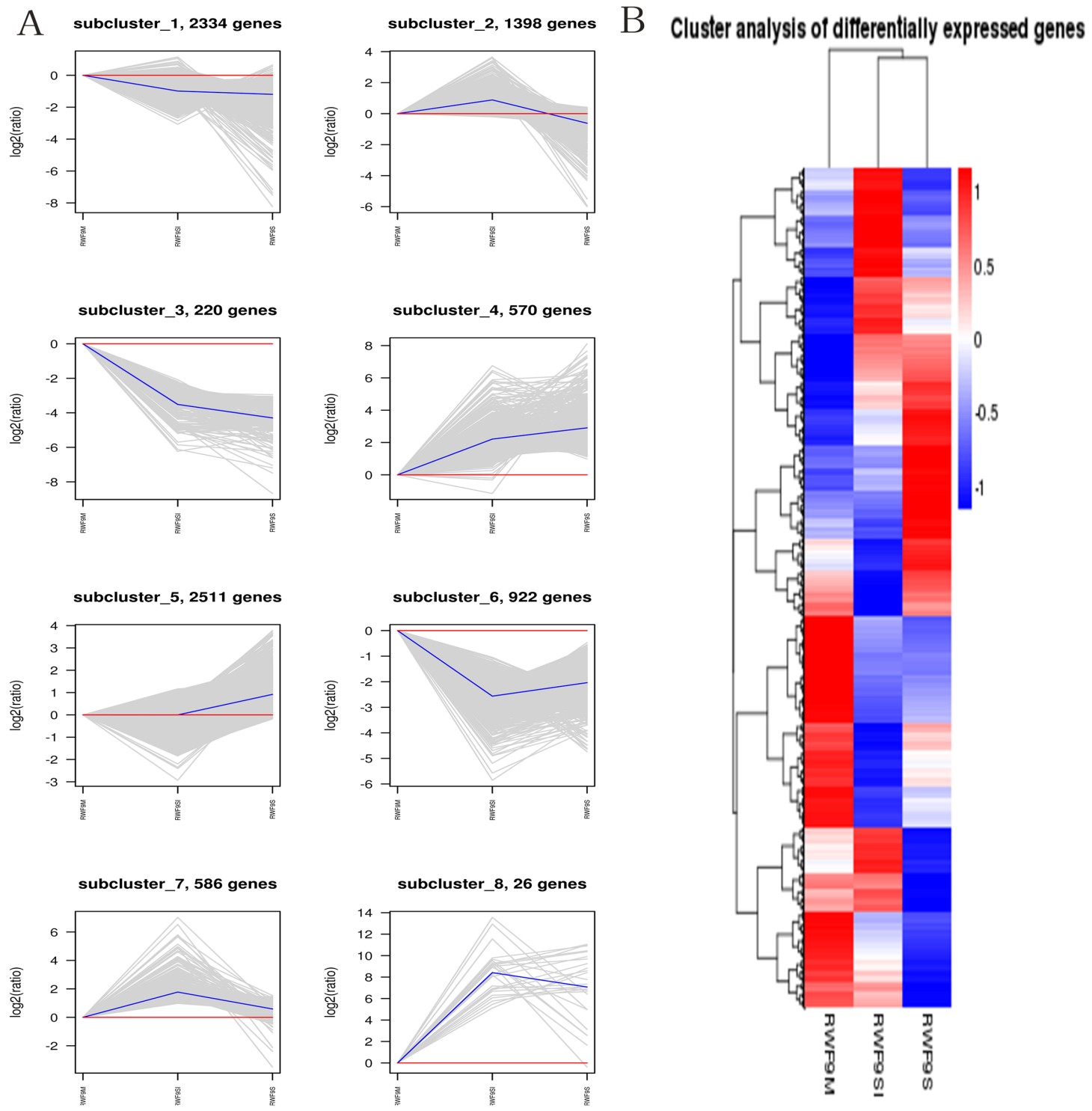

**Figure 5 The clusters analysis.** (A) Expression patterns of the genes in the eight main clusters, namely subcluster 1–8, corresponding to the heatmap. The gray is gene; The blue is gene tendency. (B) Hierarchical analysis and gene expression patterns of DEGs at three stages. Hierarchical cluster analysis of gene expression based on FPKM data. Low expression (blue) and high expression (red). The different stage is shown in the column and the transcriptional units in the rows. DEGs clustered in eight groups according to the similarity of their expression pattern.

570 genes with up-regulated expression in both stages. Subcluster 5 included 2,511 genes with higher expressionin sclerotial maturation than in the other two stages. Subcluster 6 included 922 genes with higher expression in the mycelial stage than in the two sclerotial stages. Subcluster 7 included 586 genes with higher expression in the sclerotial initiation stage than the other two stages. The 26 genes in subcluster 8 were up-regulated at both stages, and the expression levels in sclerotial initiation were higher than those in the mycelial stage, whereas those of the sclerotial maturation stage were lower compared to the sclerotial initiation stage. The top 20 genes from each category were selected (Tables S3–S10), indicating that NADPH dehydrogenase, cytochrome P450, oxygen-dependent choline dehydrogenase, chitin synthase, O-methylsterigmatocystin oxidoreductase, CAT, and NADPH-P450 reductase play particularly important roles in sclerotial formation. The DEGs of each category were annotated in GO terms and searched against the KEGG pathway (Figs. S1 and S2). The GO classification showed that a higher proportion of DEGs in subcluster 3, 4, and 8 are involved in oxidation-reduction process, oxidoreductase activity, and antioxidant activity. In subcluster 4, four genes (*SOD*, *CAT*, putative protein disulfide-isomerase, and glutathione peroxidase) were classified in the antioxidant activity GO term, and *SOD* and *CAT* were predicted to be localized at the peroxisome.

## Oxidative stress responses

To investigate the changes of ROS production during the formation of sclerotia, and changes of enzymes in ROS, the fluorescent dye $H_2DCFDA$ was used to measure intracellular ROS levels in the three stages of *R. solani* AG1 IA development. As shown in Fig. 6, ROS were detected in all three stages, although the intensity of fluorescence was much higher in the sclerotial initiation stage compared to the other two stages.

Catalase, SOD, and NOX are the key to the production and removal of ROS in the formation of sclerotia. Therefore, we assayed these enzymes. NOX and SOD showed an initial increase in activity followed by a decrease. NOX enzymes participates in the production of superoxide, an important precursor of ROS, and SOD can convert superoxide radicals to hydrogen peroxide. Thus, the increase of NOX and SOD enzyme activities in the sclerotial initiation showed that the ROS level improved. By contrast, CAT activity decreased in all three stages, which indicates a reduction of ROS scavenging (Fig. 6). In addition, we treated the samples with the CAT inhibitor aminotriazole along with $H_2O_2$, and monitored sclerotial formation, which showed that $H_2O_2$ accumulation in the absence of CAT led to the early differentiation of sclerotia (Figs. S3 and S4).

## Amplification specificity and efficiency

The gel agarose results showed all primer bands at a single size range of 100–300 bp, and the melting curves had a single peak (Figs. S5–S9). The PCR efficiency (E) of all the primers was 94.9–109.5%, and the correlation coefficients ($R^2$) were greater than 0.982 (Table S1). These results indicated that all primers could be used for subsequent RT-qPCR. A total of four reference genes were ranked by their average expression stability (*M*-value), as shown in Table S2. Comprehensive analysis showed that 18sRNA was more

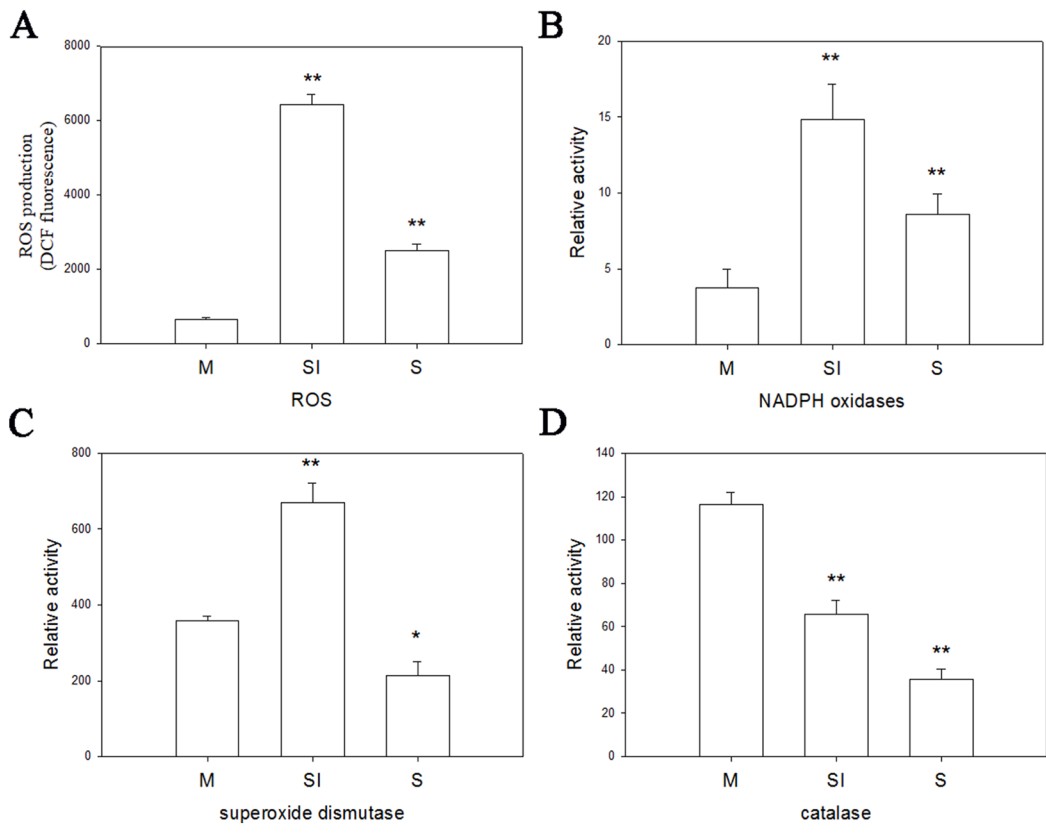

**Figure 6 Oxidative stress responses.** (A) The ROS production levels in three stages. (B) The enzyme activities of NOX1 in three stages. (C) The enzyme activities of SOD in three stages. (D) The enzyme activities of CAT in three stages. Abbreviations: M, mycelia; SI, sclerotial initiation; S, sclerotial maturation. ($P < 0.01$ *$P < 0.05$, **$P < 0.01$).

stable compared with the RNA of actin, β-tubulin, and glyceraldehyde-3-phosphate dehydrogenase.

### Validation of gene expression level changes during sclerotia formation of *R. solani* AG1 IA by RT-qPCR

The relative mRNA levels of 12 DEGs among the three stages were analyzed using RT-qPCR: 6-phosphogluconate dehydrogenase (*6PGD*), galactinol synthase 7 (*CLOS7*), glucan 1,3-beta-glucosidase, beta-galactosidase, NADH dehydrogenase, NADPH-P450 reductase, obtusifoliol 14-alpha demethylase (*CYP51*), *SOD*, *NOX*, oxygen-dependent choline dehydrogenase (*bETA*), chitin synthase D, and phosphoadenosine phosphosulfate reductase (*MET16*). The expression level trends of most of these genes were consistent between the RNA-seq and RT-qPCR results (Fig. 7).

## DISCUSSION

As far as we know, this is the first transcriptome analysis on maize sheath blight (*R. solani* AG1 IA) during the sclerotium formation process. *Kwon et al. (2014)* analyzed proteomic changes during the process of sclerotial formation of *R. solani* for 5 days, 7 days, and

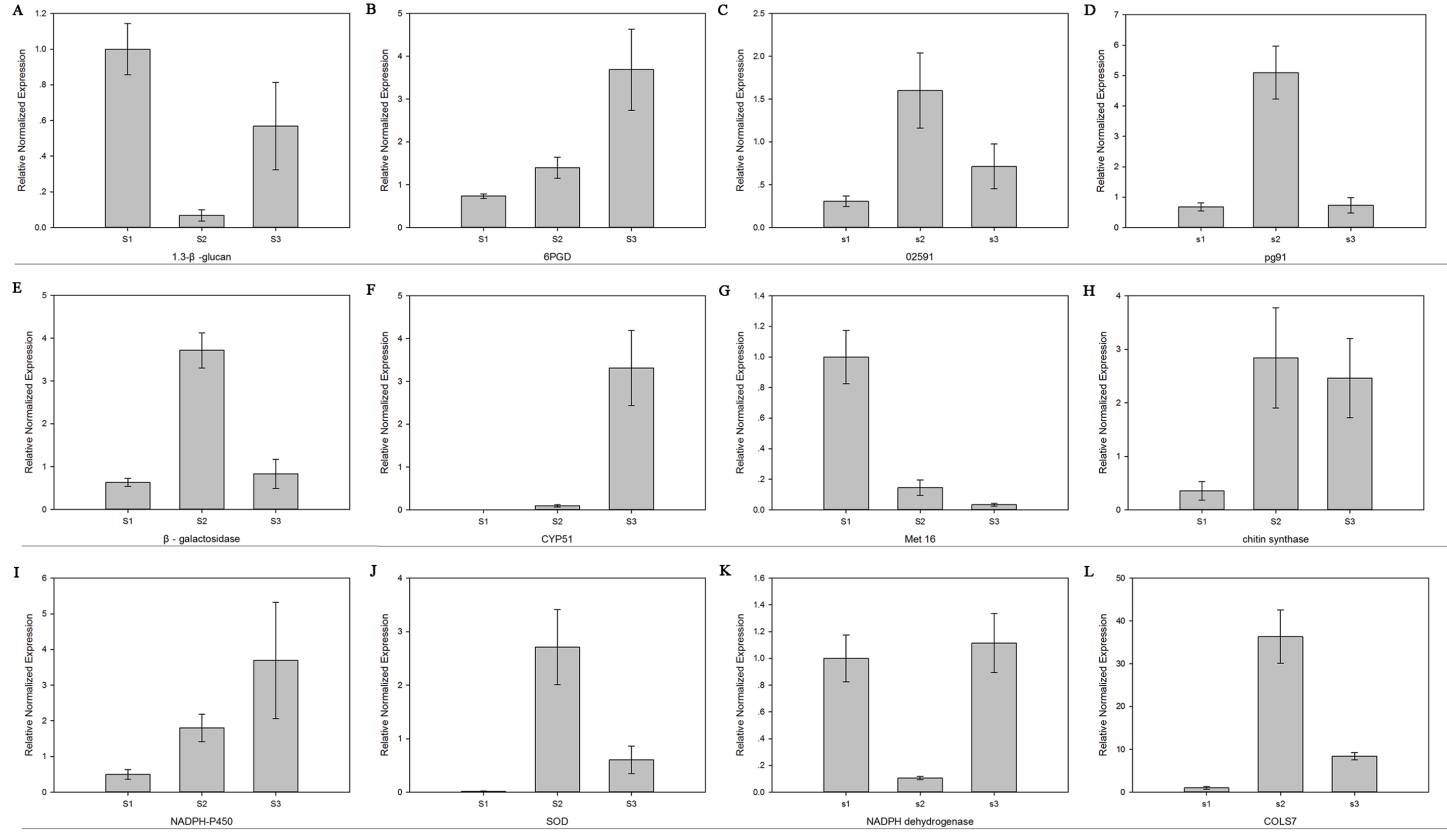

**Figure 7 The expression level of 12 genes.** Real-time quantitative PCR analysis of 12 genes in *R. solani* AG1 IA sclerotia maturation stage S1, S2, S3. (A) (1.3-β-glucan) glucan 1,3-beta-glucosidase. (B) (6PGD) 6-phosphogluconate dehydrogenase. (C) (02591) oxygen-dependent choline dehydrogenase. (D) (pg91) NADPH oxidases. (E) (β-galactosidase) beta-galactosidase. (F) (CYP51) obtusifoliol 14-alpha demethylase. (G) (Met16) phosphoadenosine phosphosulfate reductase. (H) (chitin synthase) chitin synthase D. (I) (NADPH-P450) NADPH-P450 reductase. (J) (SOD) superoxide dismutase. (K) NADH dehydrogenase. (L) (CLOS7) galactinol synthase 7. S1: RWF9M, S2: RWF9SI, S3: RWF9S.

10 days, respectively, and obtained 55 differentially expressed proteins, representing high expression of processing, cellular processes, amino acid metabolism, cell defense, and carbohydrate metabolism during the sclerotium formation. In line with these previous findings, through RNA-seq analysis during the formation of the sclerotium, we identified 5,000–6,000 DEGs between the three stages from the mycelial stage to the sclerotial initiation and maturation stages, and the majority of the highly expression were involved in the oxidation-reduction process, carbohydrate metabolic process, catalytic activity, and oxidoreductase activity. The involvement of carbohydrate metabolism suggests that carbon sources are quickly consumed in sclerotial maturation. Therefore, the cell metabolism was reduced. In addition, six antioxidant enzymes (three SODs, dihydropteroate synthase, cytochrome C peroxidase, and alpha-ketoglutarate-dependent taurinedioxygenase) of *R. solani* were differently expressed during sclerotial maturation (*Kwon et al., 2014*). In this study, SOD, dihydropteroate synthase, and cytochrome C peroxidase were found to be differentially expressed among all three stages, and more antioxidant genes were found to accumulate during the sclerotia formation process,

including CAT, putative protein disulfide-isomerase, and glutathione peroxidase. These results showed that SOD and CAT may regulate ROS levels to maintain cellular redox homoeostasis within cells during sclerotial maturation.

*Zheng et al. (2013)* suggested that in traditional breeding strategies, no crops have high resistance to *R. solani*. Although some resistance traits have been found in rice and maize, no resistance genes have been identified and cloned. However, these authors fully sequenced the transcriptome of *R. solani* AG1 IA, providing a molecular basis for the breeding of rice plants with resistance to the disease (*Zheng et al., 2013*). Our present work thus largely expands the available tools for breeding disease-resistant maize through generation of a transcriptome library of the sclerotial maturation of *R. solani* AG1 IA.

Some genes have previously been linked to sclerotia formation. *R. solani* with knockout G-protein α subunit showed a change in morphology, and the ability to form sclerotium was completely lost (*Charoensopharat et al., 2008*). The osmotic stress response and microsclerotia formation of *Verticillium dahliae* were regulated by the mitogen-activated protein kinase gene (VdHog1) (*Yonglin et al., 2016*). We further identified a large number genes related to sclerotia formation using RNA-seq technology. Moreover, the cluster analysis revealed that the expression pattern of DEGs in the sclerotial initiation and maturation stages differed from that in the mycelial stage. Eight clusters were plotted with their expression patterns, revealing significant difference of previously identified sclerotial formation-related genes. In particular, subcluster 4 included 570 genes that were up-regulated in both sclerotia stages, which were highly enriched in GO terms related to peroxidase activity, oxidoreductase activity, and oxidation-reduction process, and KEGG pathways of peroxisome, glutathione metabolism, ascorbate, and aldarate metabolism. This suggests that antioxidants (e.g., glutathione, vitamin E, and ascorbic acid) and enzymes (e.g., SOD, CAT) can change the balance of ROS and thus affect sclerotial formation.

Under hyperoxidant state, cells can be transformed between differentiated and undifferentiated states, and the ROS level is higher than that in antioxidation. In this state, the cells need to prevent premature entry into the hyperoxidant state. To prevent entry into this hyperoxidant state, cells via cell aggregation or fusion in the biogenesis of sclerotia, plasmodia, and sex organs (*Georgiou et al., 2006*). Moreover, we found that SOD and NOX activities were particularly high in the initiation stages, suggesting that the early developmental stage of sclerotium generates ROS in hyperoxidant state. In contrast, CAT activity decreased in the maturation stage, suggesting an increase in the consumption of oxygen molecules with a consequent blockage of oxygen molecules from entering the cells to balance the peroxide content. Thus, ROS is an important inducing factor for the growth of *R. solani* from the mycelium to sclerotium.

NADPH oxidase 1 (NOX1), SOD, CAT are considered to be involved in superoxide anion generation, fungal growth, and cell differentiation (*Peraza & Hansberg, 2002*; *Georgiou et al., 2006*). *Kim et al. (2011)* silenced NOX genes in *S. sclerotiorum, SsNOX1* and *SsNOX2*, and found that the production of ROS among the mutant strains decreased, sclerotia could not be formed, and the synthesis of oxalic acid—an important pathogenic factor—also decreased significantly. Although the pathogenicity of the mutant was also

significantly weakened, the gene silencing largely affected the formation of sclerotia rather than the pathogenicity of *S. sclerotiorum*. Furthermore, the mutation caused increased sensitivity to oxidative stress and significantly reduced the risk of disease (*Veluchamy et al., 2012*). Overall, our transcriptome analysis identified that changes in the expression of ROS-related genes, *NOX1*, *SOD*, and *CAT*, are key events during sclerotia formation and development. These findings were confirmed with RT-qPCR validation for 12 genes, including *MET16*, *6PGD*, and *CLOS7* in addition to *NOX1*, *SOD*, and *CAT*, which may be involved in the production of superoxide anions, fungal growth, and cell differentiation. Thus, this study provides molecular-level evidence that active oxygen plays a role in *R. solani* AG1 IA sclerotia maturation, offering further insight into the molecular mechanism of sclerotial development.

## CONCLUSION

In this study, we found a total of 8,567 DEGs throughout *R. solani* AG1 IA sclerotial maturation, which were largely associated with oxidoreductase activity, carbohydrate metabolic process, and oxidation-reduction process. Cluster analysis revealed that genes showing the same trend in expression changes during sclerotia formation may have the same function. A total of 12 of these DEGs were confirmed using RT-qPCR, including *NOX1*, *SOD*, and *CAT*, demonstrating a key role of oxidative stress through the production of superoxide anions, as well as general associations with fungal growth and cell differentiation. Based on our results, further studies should be performed to elucidate the specific metabolic pathways and functions during sclerotial development using modern techniques. The genes identified in this work only present the first step in revealing the underlying processes, but should provide a useful resource and targets for developing strategies toward the prevention and control of diseases caused by *R. solani* AG1 IA to improve crop productivity.

### Funding
This work was supported by the National Key Technology Research and Development Program (2016YFD0300704, 2017YFD0300704), and the National Natural Science Foundation of China (Grant No. 31751005). The funders had no role in study design, data collection and analysis, decision to publish, or preparation of the manuscript.

### Grant Disclosures
The following grant information was disclosed by the authors:
National Key Technology Research and Development Program: 2016YFD0300704, 2017YFD0300704.
National Natural Science Foundation of China: 31751005.

### Competing Interests
The authors declare that they have no competing interests.

## Author Contributions

- Bo Liu conceived and designed the experiments, performed the experiments, contributed reagents/materials/analysis tools, prepared figures and/or tables, authored or reviewed drafts of the paper.
- Haode Wang performed the experiments.
- Zhoujie Ma performed the experiments.
- Xiaotong Gai performed the experiments.
- Yanqiu Sun analyzed the data.
- Shidao He performed the experiments.
- Xian Liu conceived and designed the experiments, authored or reviewed drafts of the paper.
- Yanfeng Wang analyzed the data.
- Yuanhu Xuan authored or reviewed drafts of the paper.
- Zenggui Gao conceived and designed the experiments, authored or reviewed drafts of the paper.

## Data Availability

Liu, Bo (2018): DEGs qRT-PCR. figshare. Fileset. https://doi.org/10.6084/m9.figshare.5786586.v1.

Liu, Bo (2018): New Analysis results. figshare. Dataset. https://doi.org/10.6084/m9.figshare.5786634.v1.

## Supplemental Information

Supplemental information for this article can be found online at http://dx.doi.org/10.7717/peerj.5103#supplemental-information.

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
