# Peer review of "Transcriptomic evidence for involvement of reactive oxygen species in Rhizoctonia solani AG1 IA sclerotia maturation"

_PeerJ, doi:10.7717/peerj.5103_

## Round 0.1 · original submission · Major Revisions

When preparing the major revision please consider extensive English language corrections to greatly improve the use of language, spelling, and grammar. It is also advised to give more information on the used methods and their validation. Please also expand your findings and give lists of the top 20 genes and their significance under each category as recommended by reviewer 2.

Reviewer 1 ·

Basic reporting

no comment

Experimental design

no comment

Validity of the findings

no comment

Additional comments

1 Line80:corn sheath slight should be corn sheath blight.
2 Please explain expression pattern and its correlation for DEGs in RWF9SI, RWF9S and RWF9M with sclerotia maturation in GO and KEGG.
3 Line240: this is the first time that a survey analysis on the transcriptome has
been done on the sclerotium forming process of maize sheath blight (R. solani AG1 IA). Kwon et al. (2014) studied the proteomic differences in the process of sclerotial formation of R. solani for 5 days, 7 days, and 10 days respectively and obtained 55 differentially expressed proteins, finding that there is high expression of processing, carbohydrate metabolism, cell defense, amino acid metabolism, and cellular processes in the process of sclerotia. Please compare your data to the proteomic data, analyze the differences. Actually the results seem inconsistent with the reported.
4 ROS tests, NOX1 (NADPH oxidases), SOD(superoxide dismutase) and CAT (catalase) activities should be added in three stages.
5 Line 206-209:the conclusion lacks basis.
6 The expression pattern in FIG5 B should be explained in detail.
7 The accession number of sequence data in NCBI should be offered online.

Reviewer 2 ·

Basic reporting

While it was an interesting topic, the entire manuscript was hard to understand because it was not written in Standard English. For the entire manuscript, improvements should be made in the use of language, spelling, and grammar.

Experimental design

While the research question has been stated, very little information has been provided in the methods section. The authors have to expand the Methods section. For instance, the authors did not explain how the RNA was isolated from the cultures. For the quantitative PCR, how did they do the reverse transcription? Did they do a DNase digestion to eliminate the gDNA? What was the reason for the selection of 18S rRNA as the reference gene? Did they normalize the gene expression using only one reference gene?

Validity of the findings

While the data is very promising, the authors have not sufficiently expanded on their findings. For instance, what are the top 20 genes under each category? What is the significance of these genes?

Additional comments

Comments for PeerJ 2325

In this manuscript, the authors report the involvement of reactive oxygen in sclerotia formation in Rhizoctonia solani AG1-IA by carrying out transcriptomic analysis. It is an interesting study since sclerotia is the infectious propagule of the non-spore forming Rhizoctonia spp. The authors examined the transcriptome of the sclerotia formed by R. solani AG-1-IA. While it was an interesting topic, the entire manuscript was hard to understand because it was not written in Standard English. For the entire manuscript, improvements should be made in the use of language, spelling, and grammar. In addition, the authors have to expand the Methods section. For instance, the authors did not explain how the RNA was isolated from the cultures. For the quantitative PCR, how did they do the reverse transcription? Did they do a DNase digestion to eliminate the gDNA? What was the reason for the selection of 18S rRNA as the reference gene? Did they normalize the gene expression using only one reference gene? What are the top 20 gene under each category?
Most of the comments have been made to improve the readability of the manuscript. They are as follows:
1. Line 13: Change “soil borne” to “soil-borne”
2. Line 33: “soil-borne Basidiomycete fungus family”. Although I understand what the authors wish to state, however taxonomically speaking, it belongs to the class Agaricomycetes, and family Ceratobasidiaceae. Since this is a scientific article, terminology should be appropriate and not informal.
3. Line 37: Change “forlong” to “for long”
4. Line 42: it is not clear what the authors mean by “its pathogenic populations”. Is there a non-pathogenic population of R. solani AG1-IA?
5. Line 43: what do the authors mean by molecular genetic mechanism? Either the authors should use this phrase or the phrase “molecular basis of sclerotium formation”.
6. Lines 44-46 is unclear. What are differentiated substrates?
7. Line 90: How did the authors screen for the pathogenic strains. Did they inoculate the plants and re-isolate it?
8. Line 94: Did the authors store the cultures at -80 °C or 80 °C?
9. The authors did not explain how the RNA was isolated from the cultures. For the quantitative PCR, how did they do the reverse transcription? Did they do a DNase digestion to eliminate the gDNA?
10. Lines 130-135. What was the reference genome used? Was it another strain of AG1-IA? The authors gave the link of the NCBI website Genome but did not indicate which genome they used for the annotation purposes.
11. For the qPCR, why did the authors chose the 18S rRNA transcript for the normalization? Have the authors conducted a validation assay for each of the 12 genes? The authors need to establish that under the three conditions used for this study, the reference gene expression was also stable and appropriate. Furthermore, the problem with the 18S rRNA is the overabundance of the transcript. Did the authors dilute the sample in order to read fluorescence? If this happened, did this affect the fluorescence detection of the 12 genes of interest? Please refer to the paper by Kozera and Rapacz (https://www.ncbi.nlm.nih.gov/pmc/articles/PMC3825189/) as well as by Petriccione et al. (https://www.nature.com/articles/srep16961).
12. Lines 167-168 are unclear. What do the authors mean by quality contamination?
13. To improve readability, perhaps the authors can name the condition (mycelial, sclerotial initiation, or sclerotial) rather than call it by the name of the transcriptome. This refers to the Lines 185-229.
14. Lines 206-209: It is unclear why the authors conclude that because of the greater expression of transcripts associated with cellular protein metabolic process, carbohydrate metabolic process, and so on, active oxygen is involved.
15. Figure 5. The most interesting panel in the data set is perhaps sub cluster _6 (where genes are downregulated during formation of sclerotial initials and sclorotial formation) and sub cluster _8 (where genes are upregulated during formation of sclerotial initials and sclorotial formation). Perhaps, the authors can create a table and list the top 20 genes in each category?
16. Lines 230-238: The correct term to use for reverse transcription quantitative PCR is RT-qPCR and not qRT-PCR. Please refer to the guidelines for the minimum information for publication of quantitative real-time PCR experiments (http://clinchem.aaccjnls.org/content/55/4/611.long).
17. Line 254: Change “furthered” to “further”
18. Line 258-259: the authors indicate that eight clusters depicting expression patterns showed that sclerotial formation related genes were significantly up regulated. This is not an accurate assessment. In the study, the authors state that they examined genes with a p<0.05. However, Figure 5 shows the log2 ratio of FC. What are the p values associated with the FC? Do they all have p<0.05?
19. Line 263: Change “passway” to “pathway”
20. Lines 267-268: What does “hyperoxidant statedriven prevention by cell aggregation mean?

---

## Round 0.2 · Minor Revisions

Your study has been well designed and analysed. You addressed all the reviewers´ comments well enough and wrote a very good report.

There are minor comments from each reviewer which you should address, and in addition there is a note from staff below which you must address.

# Staff Note: Although the article is now scientifically acceptable, a staff check has shown that the current version of your manuscript contains an unacceptable level of text re-use from other sources. We appreciate that the same methods are often reused and that there are only so many ways to describe a procedure however we would request that you make efforts to re-write your Methods section (or to appropriately cite the papers where the method was first used). Under separate cover, we will email you the report showing the problems detected. #

Reviewer 1 ·

Basic reporting

No comment

Experimental design

No comment

Validity of the findings

No comment

Additional comments

Some minor edits are noted:

1.Line28-29, Gene Ontology and Kyoto Encyclopedia of Genes and Genomes analyses are abbreviated to GO and KEGG. In this part, you have made differentially expressed genes (DEGs),NADPH oxidase 1 (NOX1) and superoxide dismutase (SOD).

2.Line34, ‘sclerotial initiation and sclerotial maturation’ could be changed to ‘sclerotial initiation and maturation’.

3.Line74, However, the molecular mechanisms of sclerotia formation in R. solani remain poorly understood? Make sure the grammar.

4.Line87, hypothesis that reactive oxygen species (ROS) should be changed to ROS. You have mentioned this already.

5.Line119, sclerotia initiation should be sclerotial initiation.

Reviewer 2 ·

Basic reporting

The study has been well reported. The authors have indicated that they used Rhizoctonia solani Rhs1AP as the reference genome in Lines 153-154. The authors need to cite the publication associated with the genome sequence which is as follows.

Citation: Cubeta MA, Thomas E, Dean RA, Jabaji S, Neate SM, Tavantzis S, Toda T, Vilgalys R, Bharathan N, Fedorova-Abrams N, Pakala SB, Pakala SM, Zafar N, Joardar V, Losada L, Nierman WC. 2014. Draft genome sequence of the plant-pathogenic soil fungus Rhizoctonia solani anastomosis group 3 strain Rhs1AP. Genome Announc. 2(5):e1072-14. doi:10.1128/genomeA.01072-14.

Experimental design

Research questions have been well defined, and methods have been well described with sufficient detail.

Validity of the findings

Data is robust and statistically sound.

Additional comments

All the comments to the authors made previously have been addressed. Overall, the study has been well designed, studied, and reported.

---

## Round 0.3 · accepted · Accept

Congratulation, your manuscript has been accepted for publication.

#